# Understanding the relationships between 24-hour movement behavior, community mobility and the neighborhood built environment for healthy aging in Brazil: The EpiMove study protocol

**Viviane Nogueira de Zorzi**[1]*, **Janio Carlos Pessanha Coelho**[1], **Carla Elane Silva dos Santos**[1], **Joel de Almeida Siqueira Junior**[1], **Daniel Alexander Scheller**[2], **Eleonora d 'Orsi**[3], **Cassiano Ricardo Rech**[1]

1 Postgraduation Program in Physical Education, Federal University of Santa Catarina, Florianópolis, Santa Catarina, Brazil, 2 Department Health and Sport Sciences, TUM School of Medicine and Health, Technical University of Munich, Munich, Bavaria, Germany, 3 Postgraduation Program in Public Health, Federal University of Santa Catarina, Florianópolis, Santa Catarina, Brazil

* vivianedezorzi24@gmail.com

**Data Availability Statement:** No datasets were generated or analysed during the current study. All

## Abstract

### Background

The population is aging rapidly worldwide, impacting public health, with countries in the Global South, such as Brazil, aging faster than developed nations. The 24-hour movement behavior is crucial for healthy aging, but its relationship with the neighborhood built environment is underresearched, especially for older adults. The EpiMove Study uses accelerometers and GPS to investigate the relationships between 24-hour movement behavior, community mobility and the neighborhood built environment for healthy aging in older Brazilian adults.

### Methods

The EpiMove Study is a representative cross-sectional study of older adults aged 60 years and older from an urban area in the southern region of Brazil. It consists of two phases. Phase 1 involves conducting home interviews to gather subjective measures of the neighborhood built environment and physical activity. Phase 2 involves delivering devices to participants' homes and collecting objective data on 24-hour movement behavior via wrist-worn wGT3X-BT ActiGraph accelerometers and community-based active transportation via hip-mounted GPS Qstarz-1000XT devices. The data are collected simultaneously over seven consecutive days, along with the participants' reasons for adhering to the study protocol.

### Discussion

The EpiMove study will provide a better understanding of the relationships between the perceived neighborhood environment and 24-hour movement behaviors and community-based

relevant data from this study will be made available upon study completion.

**Funding:** Eleonora d"orsi received funding from the Conselho Nacional de Desenvolvimento Científico e Tecnológico (CNPq) of Brazil (grant numbers: 569834/2008-2, 475904/2013-3, and 408877/2021-9) and the Economic and Social Research Council of the United Kingdom (ESRC) through the multicentric project Promoting Independence in Dementia (PRIDE) (grant number: ES/L001802/2). The funders had no role in study design, data collection and analysis, decision to publish, or preparation of the manuscript.

**Competing interests:** The authors have declared that no competing interests exist.

active transportation among older adults, with a particular focus on whether environmental factors influence these behaviors, which are crucial for healthy aging. The results from the EpiMove study could offer essential evidence for developing public policies and urban interventions that support healthier and more equitable environments for aging populations, particularly in rapidly urbanizing regions.

## Background

Population aging is a global phenomenon that presents significant public health challenges due to the increased need for healthcare and the inadequacy of many urban areas in meeting the demands of the older adult population [1,2]. Currently, 13.2% of the population in Latin America is composed of older adults, and this number is expected to rise to 16.5% by 2030. In contrast, the projection is that 12% of the global population will be composed of older adults in the same year [3,4]. Countries in the Global South, such as Brazil, are experiencing faster population aging than are developed nations [5]. This highlights the need for public policies aimed at creating age-friendly environments that encourage the adoption of healthy behaviors among older adults living in Latin America, thereby promoting healthy aging [4,6].

For healthy aging, physical activity (PA), less time in sedentary behavior (SB), and better sleep quality are essential [7–10]. These behaviors are characterized as 24-hour movement behaviors thatinteract with each other over the course of a day [11,12]. The benefits of PA, adequate sleep, and limiting SB time are documented in public health guidelines [13–16], and not achieving these recommendations can increase health risks [17–19].

To assess 24-hour movement behavior, existing studies have employed various methods, including questionnaires [20] and wearable devices such as accelerometers [21]. Accelerometers, in particular, provide continuous and objective measurements, which are essential for understanding how 24-hour movement behavior guidelines are met [12,22].

In addition to accelerometry, the global positioning system (GPS) has increasingly been employed in aging research to objectively and reliably capture the mobility of older adults, including community mobility, defined as an individual's movement outside the home through both active (e.g., walking) and passive modes of transportation (e.g., driving and public transportation) to navigate their community [23–25]. This is particularly important because the built environment of a neighborhood can play a crucial role in facilitating compliance with public health guidelines for these behaviors [15,26–28]. Indeed, living in neighborhoods with adequate infrastructure for walking and proximity to parks and squares promotes active community mobility among older adults [29,30]. Conversely, the perception of high crime rates, lower traffic safety, and poor aesthetic quality can limit community mobility, increase the time spent in SB, and reduce sleep quality among older adults [31,32].

However, previous studies have often examined the relationship between the neighborhood's built environment and at least one of the behaviors that comprises24-hour movement behavior (PA, SB, or sleep), without considering all three behaviors together [21,33]. Studies that have investigated the 24-hour movement behavior as a whole have focused mostly on children and adolescents, while research involving this comprehensive perspective in older adults is scarce [34,35]. Additionally, while accelerometers and GPS hold significant potential for studying these behaviors, their combined use in older adult populations is still limited [36], and there is a need to develop standardized protocols for their administration to improve comparability between studies [37].

The Epimove study addresses these research gaps by combining objective measures of accelerometers and GPS for studying the relationships between the neighborhood built environment, 24-hour movement behavior and community mobilityamong older adults in Brazil. This study is essential for raising awareness among urban planners about the importance of public policies and urban interventions that promote environments conducive to these behaviors, which positively impact healthy aging in rapidly urbanizing regions.

## Main objectives

The general aim of this study protocol is to describe the methodology used in the EpiMove Study to investigate the relationships between 24-hour movement behavior, community mobility and the neighborhood built environment for healthy aging in older Brazilian adults.

On the basis of the main aim, different objectives are specified:

a. To describe the prevalence of older adults meeting the 24-hour movement behavior guidelines, including physical activity, sedentary behavior and sleep, by using accelerometers.

b. To describe the prevalence of community-based active transportation in older adults via GPS and examine its associations with sociodemographic variables.

c. To examine the associations between meeting the 24-hour movement behavior guidelines, the prevalence of community-based active transportation and perceptions of the neighborhood built environment in older adults.

d. To examine the reasons for adherence to the EpiMove Study protocol regarding the use of accelerometers and the GPS in older adults.

Disparities in sex, socioeconomic status, and skin color are considered in all the objectives of this study to recognize their significance as crucial indicators of inequalities in the Brazilian context. This study is expected to support equitable public policies for healthy aging.

## Methods/Design

### Study design and sample

The EpiMove Study is an observational and cross-sectional investigation involving older adults (≥60 years) living in an urban area of Florianopolis, Santa Catarina, Brazil. Located in southern Brazil, Florianópolis has a population of 537,211 residents, 17% of whom are aged 60 years or older, according to the latest census [38]. This city has a human development index (HDI) of 0.847, whereas Brazil's HDI is 0.760 [39]. Additionally, Florianópolis has a Gini index of 0.450, indicating income distribution inequality. This places it second to last among the 27 state capitals in Brazil. A lower score indicates less inequality [40]. The EpiMove Study adheres to the Declaration of Helsinki and has obtained ethical approval from the Ethics Research Committee with Human Subjects from the Federal University of Santa Catarina, Brazil (number 5.725.273 -CAAE: 63008222.6.0000.0121). Written informed consent is obtained from all participants prior to their inclusion in the study. The study used relevant items from the Strengthening the Reporting of Observational Studies in Epidemiology checklist as guidelines in this protocol (S1 Table) [41]. The study comprises two phases and four steps (Fig 1).

### Phase 1 of the EpiMove study

**Participant recruitment.**   In Phase 1, the inclusion criteria include men and women aged 60 years or more residing in an urban area in Florianopolis. The exclusion criteria include individuals living in long-term care institutions, hospitals, or prisons [42,43]. National data

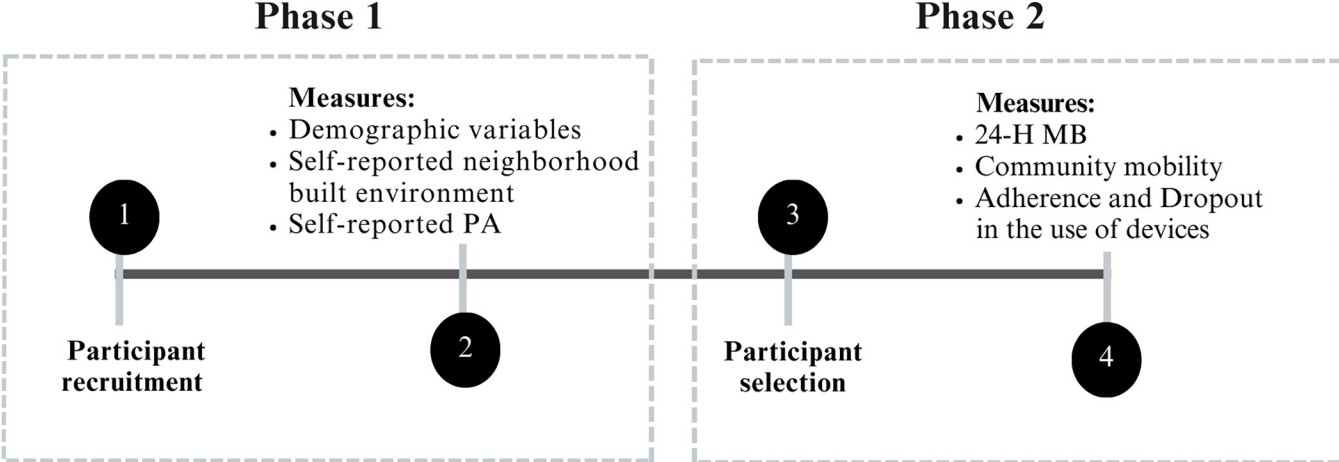

**Fig 1. Procedures in the EpiMove study.** In phase 1, older adults are recruited for home interviews to assess sociodemographic variables, self-reported physical activity, and perceptions of the neighborhood built environment. In the second phase, participants initially recruited who meet the inclusion and exclusion criteria for phase 2 will have their 24-hour movement behavior measured via accelerometers and community-based active transportation evaluated via GPS. Additionally, reasons for adherence to the EpiMove Study protocol regarding accelerometer and global positioning system use are analyzed in phase 2.

indicate that less than 1% of the older adult population resides in Long-Term Care Institutions for the Elderly and approximately 1.93% of the Brazilian prison population is composed of older adults [44,45].

The older adults invited to participate are part of the EpiFloripa Aging Cohort Study, which investigates the health conditions of this target group in Florianópolis, Santa Catarina, Brazil. This cohort study provides a representative sample of older adults in the region, ensuring that the findings are generalizable to the broader population of older residents in Florianópolis. The recruitment is conducted through a two-stage sampling process. In the first stage, urban census sectors of the municipality are selected based on the decile of income of the household head. In the second stage, households within those sectors are randomly selected.The sample size in this study is based on previous waves of the longitudinal research. In the first wave (2009/2010), 1,702 interviews were conducted with older adults from Florianópolis. In the third wave, a sample replenishment of 592 older adults was necessary to compensate for participant losses over the follow-up period. The planned sample for this protocol study will maintain this structure to ensure the representativeness of the target population and the statistical robustness needed for later analyses. Comprehensive information on the study sample and methodology of the EpiFloripa Study can be found elsewhere [43,46]. The recruitment of older adults began on May 22, 2023, for the pilot study and on March 4, 2024, for the main data collection, and is expected to continue until March 2025.

Potential participants receive an invitation letter detailing the study aims. The team members then contact the participants to schedule a home interview. If there is no response by phone, a team member visits the home to schedule the interview. Contact attempts are spaced a minimum of one week apart, alternating days and times to respect participants' routines and avoid intrusions. For example, after an attempt on Monday morning, the next attempt takes place on Tuesday afternoon of the following week. After three unsuccessful home visits, the participants' contact information are updated at the Municipal Health Secretariat of Florianópolis. Participants with updated contact information are contacted again by phone or in person. Those who cannot be located after these updates are not further considered. Refusals are recorded along with the reasons given for refusal.

**Measures.** Participants who agree to take part are scheduled for an individual face-to-face interview at their residences (Fig 1; Phase 1; Step 1). During the interview, the interviewer obtains informed consent and provides information about the study procedures. Older adults are recruited to assess sociodemographic variables, self-reported physical activity, and perceptions of the neighborhood built environment. Each interview is expected to last approximately 90 minutes.

*Sociodemographic variables*. Aspects such as age, sex, ethnicity/race, marital status, occupation, family composition and socioeconomic status are assessed through a structured questionnaire. The questionnaire includes standardized demographic questions to capture age and sex, along with self-reported data on ethnicity/race and marital status. Socioeconomic status is evaluated via indicators such as income level, education, and occupation. Family composition is assessed by inquiring about household members and their relationships with the participant (Fig 1; Phase 1; Step 2).

*Self-reported neighborhood built environment*. The perception of neighborhood built environment characteristics is assessed using an instrument adapted from the international Neighborhood Environment Walkability Scale (NEWS) [47]. The NEWS was initially validated in the U.S. by Saelens et al. (2003) and is currently internationally recommended for evaluating environmental perceptions by the International Physical Activity and Environment Network (IPEN). In Brazil, the reproducibility and validity of the translated scale have been reported [48].

This instrument consists of 34 items that assess the perceptions of the neighborhood environment characteristics of older adults in the following subscales: proximity to shops and commerce, access to services, street connectivity, walking/cycling facilities, aesthetics, traffic-related safety and crime-related safety [47,48] (S2 Table). The participants are instructed to consider their neighborhood as the area they can reach within a 15-minute walk from their residence. The questions about the presence or absence of environmental characteristics have response options where "0" represents "no" and "1" represents "yes". The items are subsequently categorized into the main dimensions, and z scores are then calculated for each dimension (Fig 1; Phase 1; Step 2).

*Self-reported physical activity*. To gain a comprehensive understanding of participants' PA levels, a validated international questionnaire (IPAQ, long form) is administered during home interviews [49]. Studies in Brazil have shown good reproducibility of the IPAQ in samples of older adults, and it has been used in population-based research investigating this age group [50,51]. The information collected covers the frequency and duration of PA performed in the leisure time and transportdomains in the past week, with a minimum duration of 10 continuous minutes per activity. The PA time is calculated by multiplying the frequency and duration of each activity and then summing these values across all activities. The IPAQ PA data is reported as minutes per day for both leisure-time PA (walking and MVPA) and transport-time PA (walking and cycling for transport). Additionally, older adults engaging in up to 149 minutes of weekly MVPA are classified as insufficiently active, whereas those reaching 150 minutes or more weekly are considered physically active (Bull et al., 2020). The total PA time (min/day), encompassing both transport and leisure-time activities, is also estimated (Fig 1; Phase 1; Step 2).

## Phase 2 of the EpiMove study

**Participant selection.** In Phase 2, additional inclusion and exclusion criteria are implemented. The inclusion criteria include older adults who have the ability to walk independently, regardless of distance, (with or without assistive devices). The exclusion criteria include older

adults whose interviews are conducted by an informant, such as a caregiver, family member, or healthcare professional, and those with severe cognitive deficits that preclude the completion of the assessments.

The participants who complete the interviews in phase 1 and meet the inclusion criteria for phase 2 are then selected and contacted by phone to participate in phase 2 of the EpiMove study. (Fig 1; Phase 2; Step 3). During this contact, participants receive detailed information about this phase and are invited to participate. If participants cannot be reached by phone after three attempts, followed by three attempts at their residence as detailed in Phase 1, they are considered lost. Those who decline participation after being informed about this phase are asked to provide the main reason for their refusal. For participants who consent to join this phase, home delivery of the accelerometer and GPS devices is scheduled.

**Measures.** In phase 2, the participants have their 24-hour movement behavior monitored via accelerometers, and community mobility is tracked via GPS technology. For GPS data collection, the sample is stratified into four classes according to walkability and household income: 1- high income and high walkability, 2- high income and low walkability, 3- low income and high walkability, and 4- low income and low walkability [52]. The goal is to reach a minimum of 60 older adults in each of these strata. Additionally, the reasons for adherence to the EpiMove Study protocol regarding the use of accelerometers and the use of a and global positioning system are investigated (Fig 1; Phase 2; Step 4).

*24-hour Movement Behavior*. The 24-hour movement behavior of older adults, including MVPA, light physical activity, SB, and sleep, is monitored via triaxial accelerometers wGT3X-BT (ActiGraph LLC, Pensacola, FL, USA). All three axes are utilized, and the sampling interval is set to 30 Hz. The devices are delivered to the participants at their homes by trained researchers, and printed usage instructions are provided. Contact information is also included for any inquiries or assistance needed (Fig 1; Phase 2, Step 4). The participants are instructed to wear the accelerometer on their nondominant wrist continuously for 24 hours a day over seven consecutive days.

Participants are provided with disposable black thin vinyl wristbands (Tag-ID, Brazil) and are instructed to remove the accelerometer only if necessary (for example, to undergo imaging exams, as well as swimming and other aquatic activities). The participants are instructed to put the accelerometers back on as soon as possible after removal. On the second and fifth days of use, participants are reminded via phone calls or messages, and any questions they ask are addressed.

The raw acceleration data from the ActiGraph devices are downloaded and converted into ".csv" files via Actilife software, version 6.12.1, for Windows. The raw data are processed in R statistical software (R Foundation for Statistical Computing) via the GGIR 3.0–5 package (http://cran.r-project.org), which considers the following: 1) autocalibration according to local gravity (America/Sao Paulo); 2) detection of extreme values and nonwear times; and 3) calculation of the mean magnitude of dynamic acceleration expressed as ENMO (Euclidean Norm Minus One) in *mg* [milligravitational units], where 1 g = 9.81 m/s$^2$ [53].

Nonwear time detection is based on the raw acceleration of the three axes via a validated algorithm [54]. The nonwear time was defined as periods of at least 60 consecutive minutes of low acceleration with little variability [54,55]. The vector magnitude of the three axes was used to calculate activity-related acceleration via the Euclidean norm minus 1 g [ENMO = $\sqrt{(x^2 + y^2 + z^2)}-1$].

Valid data are considered only when the participant wears the accelerometer and accumulates a minimum number of records ≥4 days of use (16 hours/day–GGIR standard), including one weekend day [55,56]. The full day cycle lasts from the first wake-up until the next midnight.

The data, analyzed in 5-second epochs, are used to quantify overall physical activity expressed as acceleration, as well as the time spent in SB ($\leq$18 mg), LPA (between 18 mg and 59 mg), and MPA ($\geq$60 mg), according to cut-points calibrated specifically for older adults as established in a 2021 study [57]. Sleep is calculated via a heuristic algorithm that analyzes the distribution of changes in the Z angle [58]. After data filtering, processing, exclusion due to calibration errors, and quality verification, the data are included in the database.

*Community mobility*. GPS captures participants' mobility patterns. QStarz GPS receivers, specifically the Q-1000XT model, are employed. These devices have shown robust validity and reliability and offer comprehensive geographic coordinate data (latitude and longitude), along with information on time, speed, distance, and altitude [59].

The measurement accuracy of GPS devices is established prior to data collection. First, accuracy relative to a fixed location is assessed at a geodetic point at the Federal University of Santa Catarina. The geodetic point serves as an acceptable criterion standard [60]. Establishing accuracy under static conditions is the initial priority; if the GPS receiver cannot accurately assess location when stationary, it is improbable that it will perform well under dynamic conditions. Next, the accuracy of the recording speed and movement, as well as the battery life, are evaluated. Ideally, a battery should last more than 12 hours to collect a representative daily sample. Devices that do not achieve good accuracy or the minimum required battery life are excluded from the study [60].

The selected devices are programmed with QTravel software version 1.55 to capture and store real-time data every 15 seconds and are configured with the same identification number as the accelerometer to standardize participant identification. The devices are delivered directly to participants' residences, along with chargers, a belt with a pouch, and detailed printed instructions. Contact information is also provided for any inquiries or assistance needed (Fig 1; Phase 2, Step 4). Trained researchers instruct participants to wear the GPS device on their right hip for as long as possible during the day over a period of seven consecutive days. Participants are advised to remove the GPS device only for water-based activities, such as bathing, swimming in the sea, or using a pool. The GPS devices have a rechargeable battery with a maximum duration of 36 hours; therefore, participants are instructed to remove the GPS device each night for charging and to use it again in the morning upon waking. The start buttons are sealed with adhesive tape to prevent participants from altering the data capture mode, which could compromise data acquisition. Participants using GPS receive daily messages or calls to remind them to charge the device. Additionally, they receive a daily checklist to mark each charging day, helping them remember to charge the device every night.

Data downloading is performed via QTravel software version 1.55. Device usage, including days and hours worn, is verified. The participants have to wear the GPS for at least four days: three weekdays for $\geq$10 hours/day and one weekend day for $\geq$8 hours to meet the criteria for data validation [61]. A trip is defined as a continuous period of movement exceeding 120 seconds with a distance of at least 100 meters. Table 1 below presents a detailed overview of the device specifications for data collection.

*Adherence and dropout in the use of GPS and accelerometers*. After the accelerometer and GPS data are collected, the devices are collected, and adherence and dropout questionnaires are administered to assess participant compliance with the study protocol (Fig 1; Phase 2; Step 4).

The factors influencing adherence to the accelerometer and GPS usage protocol are explored through questionnaires covering several domains: satisfaction with the device, usability difficulties, comfort, motivation, concerns about usage, impact on daily life, embarrassment, difficulties related to usability or health issues, and participant recommendations for device use. Additionally, remembering to charge the device (relevant only to the GPS) and discomfort during sleep (relevant only to accelerometers) are assessed. These aspects are

**Table 1. Summary of accelerometer and GPS device specifications for data collection.**

| | Accelerometer | GPS |
|---|---|---|
| **Device model** | ActiGraph models GT3x+ and wGT3x+ | QStarz Q-1000XT model |
| **Unit of measurement** | Raw acceleration data expressed in $mg$[a] | Latitude and longitude in degrees<br>Speed in km/h |
| **Device location** | Nondominant wrist | Right hip |
| **Sampling interval** | 30 Hz[b] | Average every 15 seconds[c] |
| **Days** | 7 consecutive days | 7 consecutive days |
| **Daytime** | Continuously for 24 hours | As long as possible during the day/removed for sleep |
| **Validation criteria** | At least 4 days:<br>3 weekdays for ≥16 hour<br>1 weekend day for ≥16 hour | At least 4 days:<br>3 weekdays for ≥10 hours/day<br>1 weekend day for ≥8 hours |
| **Cutt off point** | MVPA: ≥60 mg<br>SB: >18 mg<br>Sleep duration: Prolonged Inactivity and arm angle. | Trips with speed ≤ 10 km/h are classified as community-based active transportation. |
| **Program for initializing/downloading data** | ActiLife software version 6.13.4 | QTravel software version 1.55 |
| **Criteria for meeting guidelines** | ≥150 min/week of MVPA<br><8 h/day in SB<br>Sleep duration 7–8 h/day | N/A |

[a]Milligravitational units

[b]Data are collected 30 times per second

[c]Four measurements per minute are obtained.

evaluated via a 5-point Likert scale. The domains of "user experience with the device" and "reasons for dropout" are evaluated through open-ended questions.

## Analysis suggestion

To describe the prevalence of older adults meeting the 24-hour movement behavior guidelines, older adults will be considered to have met the 24-hour movement behavior guidelines if they meet the following criteria: spending no more than 8 hours per day in SB accumulating at least 150 minutes of MVPA over the course of the week [12,15] and sleeping between 7 and 8 hours per day [13]. Compliance with the 24-hour movement behavior guidelines will be analyzed separately for each behavior and in combination. A variable that represents how many of the 24-hour movement behavior guidelines were met (scaling from 0–3) will be created for each participant.

To describe the prevalence of community-based active transportation in older adults via a global positioning system, the analysis will be conducted as follows: Initially, trips will be categorized on the basis of speed thresholds: trips with a 90th percentile speed below 10 km/h will be classified as walking, whereas those with speeds above 10 km/h will be classified as vehicle travel [61,62]. GPS data points showing extreme speeds (>130 km/h) or significant changes in distance (>1000 m) and elevation (>100 m) between consecutive data points will be considered invalid [62,63]. The number of trips will be subsequently characterized as a dichotomous variable: the prevalence of active travel (walking) and passive travel (motorized vehicles) [62]. Additionally, the prevalence of active travel will be analyzed according to sociodemographic variables.

To examine the associations between meeting the 24-hour movement behavior guidelines, the prevalence of community-based active transportation, and the perception of the

neighborhood built environment in older adults, a multilevel binary logistic regression analysis will be conducted to estimate the odds of meeting different 24-hour movement behavior guidelines and their combinations on the basis of participants' perceptions of neighborhood environmental characteristics. Two distinct models will be employed: one will calculate the odds ratios (ORs) for meeting one, any two, or all three guidelines, and the other will calculate ORs for all possible combinations of the guidelines. Participants who do not meet any of the guidelines will serve as the reference group. Additionally, multilevel logistic regression analyses, considering the residential census tract, will be conducted to determine which perceived neighborhood environmental characteristics are significantly associated with community-based active transportation.

An adherent participant will be defined as one who wears the accelerometer for 7 valid days during the study period [64]. Dropout will be assessed by the proportion of participants who discontinued device use before the study period ends. Additionally, a valid participant will be defined on the basis of the previously specified [65].

A descriptive analysis will be conducted to examine the average number of days of device usage, average daily usage time, and proportions of valid, adherent, and dropout older adults. Furthermore, the absolute and relative frequencies of factors (satisfaction, usability, comfort, motivation, concerns about usage, impact on daily life, embarrassment, etc.) related to adherence to and dropout from the protocol will be examined and compared across sociodemographic variables. Wilcoxon rank-sum tests will compare Likert scale responses between two groups (sex), whereas Kruskal–Wallis tests will compare Likert scale responses across multiple groups (age group, education level, income, and skin color).

Additionally, reasons for dropout will be categorized and analyzed to identify common issues or barriers affecting consistent device usage. Finally, logistic regression models will be developed with "valid participants" and "adherent participants" as dichotomous dependent variables, and factors associated with device usage will be included as independent predictors.

Logistic regression parameters will be reported as OR with 95% confidence intervals (CIs), and a significance level of 5% (alpha = 0.05) will be used for all statistical procedures. The regression models will be adjusted for potential confounders to ensure the validity of the associations. The analysis will be performed via Stata 14.0 software (StataCorp, College Station, TX, USA).

## Pilot study

To ensure the successful implementation of the EpiMove study, a pilot study assessed the feasibility and acceptability of the study protocol, focusing on recruitment and data collection methods for phase 2. The pilot study included 9 female participants with an average age of 66.9 years from the Open University for Older People (UNAPI/NETI), all of whom provided informed consent. These participants had valid accelerometer and GPS data and adhered to the protocol. On the basis of the pilot study findings, the adherence and dropout questionnaires were refined to clarify the assessed dimensions. Additionally, the frequency of GPS monitoring was increased from twice a week to daily to remind participants to charge the devices nightly.

## Dissemination of results

The results will be disseminated as follows: (1) to the participants (each participant in the study will receive feedback on their 24-hour movement behavior in the form of graphical sheets with explanatory comments); (2) they will be presented at national and international conferences (partial results will be presented by the study investigators at national and

international conferences in Portuguese and English); and (3) they will be disseminated through peer-reviewed publications.

## Discussion

The EpiMove study assesses the relationships between neighborhood built environment characteristics, objectively measured 24-hour movement behavior and community mobility in older people in the context of Brazil. The use of objective measures, through accelerometers and GPS, enables a robust analysis of the movement behavior and community-based active transportation of older adults. By combining these objective measures with subjective assessments of perceived neighborhood characteristics through the NEWS, features of the built environment that either facilitate or hinder active transportation and compliance with PA, SB, and sleep guideline recommendations can be identified.

The methodology is aligned with recent trends in research. For example, the NEWS stands out as a tool that collects subjective data to assess the neighborhood environment [48,66]. The questionnaire used measures the following dimensions: proximity to shops and commercial areas, access to services, street connectivity, walking/cycling facilities, aesthetics, traffic-related safety, and crime-related safety [47,48], making it a comprehensive measure with the potential to compare study results across different locations [67,68].

For collecting data via accelerometers, wrist-worn accelerometers were chosen in contrast to hip-worn devices, as they provide more accurate data on sleep duration and efficiency, which are fundamental aspects of older adults' health [57,69]. Additionally, wrist-worn accelerometers demonstrate greater compliance and acceptability than hip-worn devices do [70,71]. Another important aspect of the protocol is the use of waterproof disposable wristbands for the accelerometers. Compared with reusable fabric wristbands, this type of wristband can increase the wearing time of the accelerometer and facilitate greater compliance with usage time criteria in adolescents [72]. However, it remains unclear whether this effect applies equally to older adults [73]. The hypothesis is that, being waterproof, the wristbands do not need to be removed during bathing or daily activities such as washing dishes and clothes. Consequently, the possibility of forgetting them and the need to put them back on is reduced, which may increase adherence to the accelerometer usage protocol.

Regarding GPS data collection, two points need to be considered. First, the accuracy of GPS data can be affected by environmental factors such as tall buildings, densely wooded areas, or adverse weather conditions, which can obstruct or reflect satellite signals. Second, another challenge is the need to charge GPS devices every night, which can result in incomplete data due to forgetfulness or a lack of adherence. Indeed, declines in processing speed and memory occur with aging [25,73]. This underscores the need to develop adapted strategies to improve adherence to protocols tailored to the characteristics of the study population [74,75].

A combination of investigator-initiated and participant-based strategies is recommended to promote protocol adherence [60,76]. Investigator-initiated strategies involve activities initiated by researchers (e.g., follow-up calls to participants during the measurement period) to assist participants in successfully following the protocols, whereas participant-based strategies include additional tasks (e.g., activity diaries) designed to promote adherence [73,76]. Several strategies were employed to address this last challenge: 1) clear and user-friendly presentation of device usage instructions; 2) a checklist for participants to mark each charging session; 3) daily monitoring calls and messages; and 4) immediate contact in the case of questions.

Adherence to wearable devices is crucial for interpreting the effects of observational studies, yet it is often unreported [73]. Given the specific needs of older adults and the context of Latin America, enhancing implementation strategies for these devices is essential. This involves

identifying factors that may facilitate or hinder their use [73,77,78]. The study assesses various influencing factors, including sociodemographic characteristics, satisfaction, usability, comfort, motivation, impact on daily life, and feelings of embarrassment. Understanding these factors is key to increasing participation among older adults and reducing dropout rates, thereby ensuring a more representative sample [73,74].

To ensure the effectiveness of the protocol and improve the overall process, a pilot testing phase was conducted. This preliminary phase was essential in refining both the monitoring protocols and the strategies for home visits. Pilot testing revealed that the use of phone calls and text messages for scheduling device deliveries and sending reminder messages was highly effective.

Where people live determines healthy aging through barriers or incentives that affect opportunities, experiences, decisions, and behaviors in old age [4,79]. The EpiMove protocol study advances the understanding of how neighborhood environments influence 24-hour movement behaviors and community mobility among older adults, aligning with the WHO's Decade of Healthy Aging, which emphasizes creating environments that promote healthy aging [80]. It also aligns with the Sustainable Development Goals (SDGs), particularly with respect to health and well-being (SDG 3), reducing inequalities (SDG 10), and developing sustainable cities and communities (SDG 11) [81]. Given the rapid growth of the older adult population in Brazil and other Global South countries, this study underscores the need for robust data to guide public policies and urban interventions that promote healthier and more equitable environments. However, it is important to consider that Florianópolis has a relatively higher HDI compared to the national average, which may limit the generalizability of future results to other regions with different socio-economic conditions. Furthermore, the city's Gini index indicates a high degree of income inequality, reflecting disparities in quality of life and healthcare access among older adults. These factors should be considered when interpreting the results, as they provide a broader context for understanding the living conditions within the city.

## Supporting information

**S1 Table. STROBE statement.** Checklist of items that should be included in reports of cross-sectional studies.
(DOCX)

**S2 Table. Subscales and sample items from the neighborhood environment walkability scale.**
(DOCX)

## Acknowledgments

Thanks to Drs. Adriano Akira Hino, Kelly Samara da Silva, Rogério Fermino, Marcus Lopes, and Tania Bertoldo Benedetti for their essential contributions with suggestions and materials for the EpiMove Study. The students Thamires Coco, João Vítor Santos da Silva, Rafaella Oliveira, and Júlio César Padilha are also acknowledged for their fundamental collaboration in the progress of the study.

## Author Contributions

**Conceptualization:** Viviane Nogueira de Zorzi, Cassiano Ricardo Rech.

**Methodology:** Viviane Nogueira de Zorzi, Cassiano Ricardo Rech.

**Project administration:** Eleonora d 'Orsi.

**Resources:** Eleonora d 'Orsi.

**Supervision:** Cassiano Ricardo Rech.

**Visualization:** Viviane Nogueira de Zorzi.

**Writing – original draft:** Viviane Nogueira de Zorzi, Janio Carlos Pessanha Coelho, Carla Elane Silva dos Santos, Joel de Almeida Siqueira Junior, Daniel Alexander Scheller.

**Writing – review & editing:** Viviane Nogueira de Zorzi.

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
