## [Decision Letter · Decision Letter 0]

9 Oct 2024

PONE-D-24-37986Understanding the relationships among 24-hour movement behavior, active commuting and the neighborhood built environment for healthy aging in Brazil: The EpiMove Study protocolPLOS ONE

Dear Dr. Zorzi,

Thank you for submitting your manuscript to PLOS ONE. After careful consideration, we feel that it has merit but does not fully meet PLOS ONE’s publication criteria as it currently stands. Therefore, we invite you to submit a revised version of the manuscript that addresses the points raised during the review process.

**ACADEMIC EDITOR: Dear Author, please attend to all the comments provided by the reviewers and make necessary corrections.**

We look forward to receiving your revised manuscript.

Kind regards,

Zulkarnain Jaafar

Academic Editor

PLOS ONE

Journal Requirements:

“Eleonora d"orsi received funding from the Conselho Nacional de Desenvolvimento Científico e Tecnológico (CNPq) of Brazil (grant numbers: 569834/2008-2, 475904/2013-3, and 408877/2021-9) and the Economic and Social Research Council of the United Kingdom (ESRC) through the multicentric project Promoting Independence in Dementia (PRIDE) (grant number: ES/L001802/2).”

Reviewers' comments:

Reviewer's Responses to Questions

**Comments to the Author**

1. Does the manuscript provide a valid rationale for the proposed study, with clearly identified and justified research questions?

Reviewer #1: Yes

Reviewer #2: Yes

2. Is the protocol technically sound and planned in a manner that will lead to a meaningful outcome and allow testing the stated hypotheses?

Reviewer #1: Yes

Reviewer #2: Yes

3. Is the methodology feasible and described in sufficient detail to allow the work to be replicable?

Reviewer #1: Yes

Reviewer #2: Yes

4. Have the authors described where all data underlying the findings will be made available when the study is complete?

Reviewer #1: No

Reviewer #2: Yes

5. Is the manuscript presented in an intelligible fashion and written in standard English?

Reviewer #1: Yes

Reviewer #2: No

6. Review Comments to the Author

You may also provide optional suggestions and comments to authors that they might find helpful in planning their study.

Reviewer #1: TO THE AUTHORS

Title: I recommend changing the title for grammatical clarity from “the relationships among...” to “the relationships between...”.

Background:

• It is unclear why commuting is being quantified in a population aged 60+. Many participants may be retired or close to retirement, making commuting less relevant. If you meant mobility within the local community, please rephrase both the relevant sentences in the manuscript and the study title.

• Line 34: The statement is unclear. What is the projected 12% meant to indicate?

• Line 43: Physical activity advice now emphasizes that "any amount of physical activity is important," rather than just 150 minutes per week. Please refer to the UK Chief Medical Officers' recommendations for older adults, particularly based on the study by Onambele-Pearson et al. (2019) (Front Physiol 2019;10:408).

Methods

• Line 104: An HDI of 0.847 (Very High Human Development) is significantly greater than 0.760 (High Human Development). Although the difference seems small (0.087), the HDI scale is sensitive, indicating meaningful disparities in quality of life, healthcare access, and overall well-being. This could affect the generalizability of your results, which should be highlighted in your reports.

• Line 105: The Gini index of 0.450 suggests notable income inequality in your region of study. Ensure broad recruitment across socio-economic strata to maintain the representativeness of your sample.

• Line 125: It would be helpful to include statistics on the percentage of the older population in your region living in long-term care institutions, hospitals, or prisons.

• Lines 132-133: Please provide a brief explanation of your sample size justification.

• Line 142: Five attempts to contact participants at Phase 1 (plus six in Phase 2, Line 202) seems excessive and may create undue pressure. Could this be considered borderline harassment? To mitigate this:

o A) Specify how contact attempts will be spaced out (e.g., days/weeks between attempts).

o B) Have participants previously agreed to multiple contact attempts?

o C) Consider reducing in-person visits (3 seems intrusive); a mix of phone, email, and one in-person visit may be less burdensome.

o D) Ensure that participants fully understand why they are being contacted and what their participation involves.

• Lines 145-148: Ethical best practices recommend providing clear study information in advance, giving participants time to review it, and allowing them to ask questions before consenting. Immediate consent, especially in their homes, may create discomfort and pressure. This is not reflected in your current approach, which raises concerns.

• Lines 175-189: The rationale for using the IPAQ is unclear when:

o a) Accelerometers will provide more reliable physical activity data.

o b) The IPAQ adds to participant burden by extending the interview.

o c) The IPAQ does not effectively capture sedentary behaviour, which is a key aspect of the physical activity spectrum (Ryan et al., PLoS One 2018;13). I recommend reconsidering the use of the IPAQ.

• Line 193: Please specify the time or distance for the walking ability test (e.g., walk for 2 minutes? Walk 30 meters?).

• Lines 249-254: Provide details on the origin of the cut-off values, including whether they were calibrated in a similar population.

Reviewer #2: The explanation of the research in the introduction is quite weak and does not adequately explain the characteristics and reasons for acquiring them. Need more references for citing in the discussion. The text contains many grammatical errors. Language editing is essential for resubmission.

7. PLOS authors have the option to publish the peer review history of their article (what does this mean?). If published, this will include your full peer review and any attached files.

Reviewer #1: No

Reviewer #2: **Yes: **DR. YAHYA H. Y. ALFARRA, BDS (Hons), MSc, PhD

---

## [Author Response · Author response to Decision Letter 0]

8 Nov 2024

Reviewer #1

Title:

1. I recommend changing the title for grammatical clarity from “the relationships among...” to “the relationships between...”.

Response: Thank you for the suggestion! Changing the title from "the relationships among..." to "the relationships between..." will enhance grammatical clarity. 

Background:

1. It is unclear why commuting is being quantified in a population aged 60+. Many participants may be retired or close to retirement, making commuting less relevant. If you meant mobility within the local community, please rephrase both the relevant sentences in the manuscript and the study title.

Response: We appreciate your observation regarding the term "active commuting" and agree that "mobility within the local community" more accurately describes the focus of our study. In response, we defined this term in the introduction and made changes to the title and body of the text to reflect this terminology.

Although approximately 80% of our sample is retired, the concept of "community mobility" encompasses an individual's movement outside the home, including both active modes of transportation (such as walking) and passive modes (such as driving and using public transportation) to navigate within the community. This concept is essential for promoting health among older adults, regardless of their occupational status. In the Brazilian context, where access to public transportation is limited in some regions, many older adults still rely on active commuting as their primary means of mobility for daily activities, often out of necessity. Investigating how the built environment facilitates or hinders this active mobility is crucial for developing public policies aimed at healthy aging.

2. Line 34: The statement is unclear. What is the projected 12% meant to indicate?

Response: Thank you for your feedback. The sentence highlights the increasing proportion of older adults in the population of Latin America compared to the global projection for the same age group. Currently, 13.2% of the population in Latin America is composed of older adults, and this percentage is expected to rise to 16.5% by 2030. This indicates that while Latin America is experiencing a more pronounced aging trend, the global average for older adults is projected to be only 12% in 2030. This comparison underscores the significance of the aging population in Latin America. The text has been modified for better clarity.

3. Line 43: Physical activity advice now emphasizes that "any amount of physical activity is important," rather than just 150 minutes per week. Please refer to the UK Chief Medical Officers' recommendations for older adults, particularly based on the study by Onambele-Pearson et al. (2019) (Front Physiol 2019;10:408).

Response: Thank you for your comment and suggestion. We agree that the guidance on physical activity should emphasize that "any amount of physical activity is important." Our guidelines also include similar recommendations. We recognize the importance of reviewing this section and have made the necessary changes in the text to reflect these guidelines.

Methods:

1. Line 104: An HDI of 0.847 (Very High Human Development) is significantly greater than 0.760 (High Human Development). Although the difference seems small (0.087), the HDI scale is sensitive, indicating meaningful disparities in quality of life, healthcare access, and overall well-being. This could affect the generalizability of your results, which should be highlighted in your reports.

Response: We appreciate your observation regarding the difference in the Human Development Index (HDI) between the city of Florianópolis (0.847) and Brazil as a whole (0.760). We agree that this difference is significant and may limit the direct generalization of our findings to all regions of the country. However, we would like to emphasize that the Gini index for Florianópolis (0.450) reflects a marked income distribution inequality within the city itself, which somewhat mirrors the disparities present throughout Brazil. This inequality can impact the quality of life and access to healthcare services among older adults, which should also be considered when interpreting the results of our study. Therefore, the combination of these indices provides a more comprehensive view of living conditions in Florianópolis. We have incorporated this limitation at the end of the discussion section to highlight it in the manuscript.

2. Line 105: The Gini index of 0.450 suggests notable income inequality in your region of study. Ensure broad recruitment across socio-economic strata to maintain the representativeness of your sample.

Response: We appreciate your observation regarding the Gini index of 0.450, which indicates notable income inequality in our study region. To address this issue, we would like to clarify that the recruitment was conducted through a two-stage sampling process. In the first stage, we selected 80 urban census sectors of the municipality based on the decile of income of the household head. In the second stage, we performed a random selection of households within those sectors. This approach, as described by Confortin et al. (2022) [43], aims to ensure the representativeness of our sample. This information has been included in the article for clarity.

3. Line 125: It would be helpful to include statistics on the percentage of the older population in your region living in long-term care institutions, hospitals, or prisons.

Response: Thank you for the suggestion. We have included the relevant national data on the percentage of older adults residing in Long-Term Care Institutions for the Elderly and in prisons in the Methods section to contextualize our exclusion criteria. Specifically, a survey in Brazil has shown that less than 1% of the older adult population resides in Long-Term Care Institutions for the Elderly (ILPIs) (Camarano & Kanso, 2010), and another study indicated that 1.93% of the Brazilian prison population is composed of older adults (Brazil, 2023); however, we lack regional data (Pollo & Assis, 2020). Our search in studies and government databases did not yield public information on the percentage of elderly individuals in hospitals, either nationally or regionally. We requested this data from the Public Prosecutor's Office but, to date, have not received a response within the necessary timeframe.

Camarano, Ana Amélia, and Solange Kanso. "As instituições de longa permanência para idosos no Brasil." Revista brasileira de estudos de população 27 (2010): 232-235. 

Pollo, Sandra Helena Lima, and Mônica de Assis. "Instituições de longa permanência para idosos-ILPIS: desafios e alternativas no município do Rio de Janeiro." Revista Brasileira de Geriatria e Gerontologia 11 (2019): 29-44.

Brazil. Custodial Procedures for Elderly People in the Prison System. National Secretariat of Penal Policies. Brasília: Ministry of Justice and Public Security, 2023. 24 p.

4. Lines 132-133: Please provide a brief explanation of your sample size justification.

Response: Thank you for your comment. The justification for the sample size in this study is based on previous waves of the longitudinal research. In the first wave (2009/2010), 1,702 interviews were conducted with older adults. In the third wave, a sample replenishment of 592 older adults was carried out to compensate for participant losses over the follow-up period. The planned sample for this protocol study maintains this structure to ensure the representativeness of the target population and the statistical robustness needed for longitudinal analyses. We will adjust the text to improve clarity and understanding.

5. Line 142: Five attempts to contact participants at Phase 1 (plus six in Phase 2, Line 202) seems excessive and may create undue pressure. Could this be considered borderline harassment? To mitigate this:

Response: We appreciate the detailed comments. We believe that respectful interaction and clear communication are essential, especially when working with an older population that may face access and comprehension challenges related to technology. Participants are repeatedly reminded that participation in the project is voluntary and that they may withdraw at any stage without any consequences. The contact attempts follow a carefully planned approach to minimize any discomfort, based on the specific needs of this population and on ethical research practices.

A) Specify how contact attempts will be spaced out (e.g., days/weeks between attempts).

Response: Contact attempts are spaced a minimum of one week apart, alternating days and times to respect participants’ routines and avoid intrusions. For example, after an attempt on Monday morning, the next attempt would take place on Tuesday afternoon of the following week. This spacing, which has been clarified in the manuscript, aims to increase the likelihood of effective contact without pressuring the participant, keeping attempts to the minimum necessary to achieve a representative sample.

B) Have participants previously agreed to multiple contact attempts?

Response: In the first phase, participants are informed about the possibility of multiple contacts throughout the study. Only three interactions are considered essential and effective: the initial interview and the visits for device setup and retrieval in the second phase. The nature and frequency of the contacts are explained from the start, and participants have the freedom to accept or decline each step, ensuring complete autonomy.

C) Consider reducing in-person visits (3 seems intrusive); a mix of phone, email, and one in-person visit may be less burdensome.

Response: Due to the participants’ age and education profile (with approximately 32% having 4 years of education or less), the three in-person visits are necessary to ensure instruction clarity and correct device usage. The visits for device setup and retrieval are brief, lasting between five and 10 minutes, and aim to instruct and answer questions regarding device usage. Practical experience indicates that in-person communication reduces uncertainty and increases data reliability, ensuring that devices are used correctly.

D) Ensure that participants fully understand why they are being contacted and what their participation involves.

Response: Transparency is prioritized in each interaction. Participants receive a detailed invitation letter before the study, and during the first interview, they receive a full explanation of the study phases and the purpose of each visit. We reinforce this information in each subsequent contact to ensure they understand the value of their participation and what each study stage entails. These measures aim to ensure fully informed consent and that participants feel comfortable to participate or decline without any pressure. 

We hope these clarifications address your concerns, and we are available to discuss any additional adjustments necessary to ensure maximum respect and comfort for participants during data collection.

6. Lines 145-148: Ethical best practices recommend providing clear study information in advance, giving participants time to review it, and allowing them to ask questions before consenting. Immediate consent, especially in their homes, may create discomfort and pressure. This is not reflected in your current approach, which raises concerns.

Response: We appreciate the opportunity to clarify our consent and recruitment process. Our approach follows rigorous ethical practices, considering both the longitudinal nature of the study and the specific vulnerabilities and needs of the older population. Here are some important points:

A) This study includes participants involved since 2009 or 2015, who are familiar with the study’s goals and procedures. All participants receive a detailed invitation letter before any contact, explaining the study's objectives, methods, and expectations. This provides ample time to consider participation. Those who are not interested may decline participation upon receiving the letter (which includes our contact information for follow-up) or during the phone contact, without any need for a home visit. If they decline or wish to withdraw at any stage, we respect their decision without further contact attempts. Any refusal or withdrawal is respected without additional contact attempts.

B) Only participants who agree to participate or whom we have been unable to contact by phone—due to outdated contact information, lack of a phone, or other limitations (including death, which is more common in this age group, especially following the pandemic)—are visited at home. In Brazil, especially in this age group, access to technology can be challenging, and in some cases, a home visit becomes the only way to confirm availability to participate. During the visit, participants receive explanations and have the opportunity to ask questions (noting that they have participated at least once before). If they decline, we respect their right to immediately refuse without further contact attempts. Any refusal, at any stage, is respected without additional contact attempts.

C) Recognizing the challenges many participants face with technology, we are initiating an innovative project called Conecta EpiFloripa to train older adults in basic smartphone functions, such as sending emails, messages, and making and receiving calls. This parallel initiative aims to facilitate contact and make the study more accessible, enhancing participants' digital inclusion.

We hope this information addresses your concerns, ensuring that our approach fully respects participants' autonomy and comfort while facilitating participation in an ethical and transparent manner.

7. Lines 175-189: The rationale for using the IPAQ is unclear when:

a) Accelerometers will provide more reliable physical activity data.

b) The IPAQ adds to participant burden by extending the interview.

c) The IPAQ does not effectively capture sedentary behaviour, which is a key aspect of the physical activity spectrum (Ryan et al., PLoS One 2018;13). I recommend reconsidering the use of the IPAQ.

Response: Thank you for the feedback on the use of the IPAQ. While accelerometers provide reliable objective data on physical activity, sedentary behavior, and sleep, the IPAQ adds essential context by capturing the domain of physical activity—whether driven by necessity or choice. Studies such as Salvo et al. (2023) emphasize that understanding this distinction is fundamental for promoting more equitable aging. This distinction is particularly critical, as older adults in low- and middle-income contexts often engage in physical activities not as a health choice, but due to environmental or economic needs (e.g., commuting to access services).

Capturing this information is particularly relevant in the context of the older population in Brazil, where disparities in activity domains often reflect and amplify existing health inequalities. Additionally, we recognize the additional time burden that the IPAQ may impose. To mitigate this impact, our interviewers are extensively trained and use optimized tablet-based tools, which minimize participants' time and improve the efficiency of the data collection process. 

Salvo, D., Jáuregui, A., Adlakha, D., Sarmiento, O. L., & Reis, R. S. (2023). When moving is the only option: the role of necessity versus choice for understanding and promoting physical activity in low-and middle-income countries. Annual Review of Public Health, 44(1), 151-169.

8. Line 193: Please specify the time or distance for the walking ability test (e.g., walk for 2 minutes? Walk 30 meters?)

Response: Thank you for the insightful comment. In Phase 2, the inclusion criterion of “independent walking ability” refers to participants' capacity to move independently without human assistance, regardless of distance, while allowing the use of assistive devices such as canes or walkers. This criterion aligns with recommendations in the literature, such as in Studenski et al. (2011), which emphasize the importance of functional autonomy for older adults, even when assistive devices are required. 

Studenski, S., Perera, S., Patel, K., Rosano, C., Faulkner, K., Inzitari, M., ... & Guralnik, J. (2011). Gait

---

## [Decision Letter · Decision Letter 1]

20 Nov 2024

Understanding the relationships between 24-hour movement behavior, community mobility  and the neighborhood built environment for healthy aging in Brazil: The EpiMove Study protocol

PONE-D-24-37986R1

Dear Dr. de Zorzi,

We’re pleased to inform you that your manuscript has been judged scientifically suitable for publication and will be formally accepted for publication once it meets all outstanding technical requirements.

Kind regards,

Zulkarnain Jaafar

Academic Editor

PLOS ONE

Additional Editor Comments (optional):

Reviewers' comments:

Reviewer's Responses to Questions

**Comments to the Author**

1. Does the manuscript provide a valid rationale for the proposed study, with clearly identified and justified research questions?

Reviewer #1: Yes

Reviewer #2: Yes

2. Is the protocol technically sound and planned in a manner that will lead to a meaningful outcome and allow testing the stated hypotheses?

Reviewer #1: Yes

Reviewer #2: Yes

3. Is the methodology feasible and described in sufficient detail to allow the work to be replicable?

Reviewer #1: Yes

Reviewer #2: Yes

4. Have the authors described where all data underlying the findings will be made available when the study is complete?

Reviewer #1: Yes

Reviewer #2: Yes

5. Is the manuscript presented in an intelligible fashion and written in standard English?

Reviewer #1: Yes

Reviewer #2: Yes

6. Review Comments to the Author

You may also provide optional suggestions and comments to authors that they might find helpful in planning their study.

Reviewer #1: All comments have been addressed by the authors to the satisfaction of this reviewer. No further comment remain

Reviewer #2: The authors amended the original manuscript to incorporate the recommendations of the reviewers. Therefore, I have no further suggestions for this article.

7. PLOS authors have the option to publish the peer review history of their article (what does this mean?). If published, this will include your full peer review and any attached files.

Reviewer #1: No

Reviewer #2: **Yes: **DR. YAHYA H. Y. ALFARRA, BDS (Hons), MSc, PhD

---

## [Editor Report · Acceptance letter]

25 Nov 2024

PONE-D-24-37986R1 

PLOS ONE

Dear Dr. Zorzi, 

I'm pleased to inform you that your manuscript has been deemed suitable for publication in PLOS ONE. Congratulations! Your manuscript is now being handed over to our production team.

Kind regards, 

on behalf of

Dr. Zulkarnain Jaafar 

Academic Editor

PLOS ONE